# Efficient simulation of non-Markovian dynamics on complex networks

Gerrit Großmann[1]*, Luca Bortolussi[1,2], Verena Wolf[1]

**1** Saarland Informatics Campus, Saarland University, Saarbrücken, Germany, **2** Department of Mathematics and Geosciences, University of Trieste, Trieste, Italy

* gerrit.grossmann@uni-saarland.de

**Data Availability Statement:** We only use synthetically generated data. The process is described in the manuscript. Code is available on Github at https://github.com/gerritgr/non-markovian-simulation.

## Abstract

We study continuous-time multi-agent models, where agents interact according to a network topology. At any point in time, each agent occupies a specific local node state. Agents change their state at random through interactions with neighboring agents. The time until a transition happens can follow an arbitrary probability density. Stochastic (Monte-Carlo) simulations are often the preferred—sometimes the only feasible—approach to study the complex emerging dynamical patterns of such systems. However, each simulation run comes with high computational costs mostly due to updating the instantaneous rates of interconnected agents after each transition. This work proposes a stochastic rejection-based, event-driven simulation algorithm that scales extremely well with the size and connectivity of the underlying contact network and produces statistically correct samples. We demonstrate the effectiveness of our method on different information spreading models.

## Introduction

Networks provide a general language for the representation of interconnected systems. Computational modeling of stochastic dynamical processes happening on top of network typologies is a thriving research area [1–3]. Here, we consider continuous-time spreading dynamics *on* networks. That is, at each point in time all agents (i.e. nodes) occupy a specific local state (resp. compartment, internal state, or node state). The node states change over time but the underlying *contact network* which specifies the connectivity remains the same.

The most common framework for such processes is the Susceptible—Infected—Susceptible (SIS) model and its many variants [4–6]. In the SIS model, agents are either susceptible (healthy) or infected. Infected agents can recover (become susceptible again) or infect neighboring susceptible agents. SIS-type diffusion models have proven to be useful for the analysis, prediction, and reconstruction of opinion- and meme-spreading in online social networks [7, 8] as well as for the propagation of neural activity [9, 10], and the spread of malware [11] and blackouts in financial institutions [12, 13].

Agents change their state either by interacting with another agent (e.g., they become infected) or spontaneously and independently from their neighbors (e.g., when they recover). We call the state change of an agent an *event*. Previous work focused primarily on so-called

**Funding:** This work was partly funded by the German Research Foundation (DFG) under grant SFB 1223. There was no additional external funding received for this study.

**Competing interests:** The authors have declared that no competing interests exist.

*Markovian* models, in which the probability of an agent changing its state in the next infinitesimal time unit is constant (that is, independent of the time the agent has already spent in its current state). We call these agents *memoryless*, because they don't "remember" how much time they have already spend in their internal state.

As a consequence of the memoryless property, the time until an agent changes its local state follows an exponential distribution. The exponential distribution is parameterized by some rate $\lambda \in \mathbb{R}_{\geq 0}$. This rate can vary for different types of events (recovery, infection, etc.) and depend on the direct neighborhood.

It is long known that it is unrealistic to assume exponentially distributed inter-event times in many real-world scenarios. As empirical results show, this holds for instance for the spread of epidemics [14–18], opinions in online social networks [19, 20], and neural spike trains [21]. Assuming inter-event times that can follow arbitrary distributions complicates the analysis of such processes. Often Monte-Carlo simulations are the only feasible way to investigate the emerging dynamics, but even these suffer from high computational costs. Specifically, they often scale badly with the size of the contact networks.

Recently, Masuda and Rocha introduced the Laplace-Gillespie algorithm (LGA) for the simulation of non-Markovian dynamics on networks in [22]. The method is based on an earlier approach, the non-Markovian Gillespie algorithm (nMGA) by Boguná et al [23]. Masuda and Rocha aim at minimizing the computational burden of sampling inter-event times. However, both methods, nMGA and LGA, require a computationally expensive updating of an agent's neighborhood in each simulation step. We explain both methods in more detail later. For Markovian spreading models, rejection-based simulation was recently successfully applied to overcome these limitations [24–26].

## Contribution

ChangeColor This work is an extension of [26] in terms of theoretical analysis and experimental evaluation of our method. Specifically, we provide an additional case study, add a correctness and runtime analysis, and investigate the limitations of our method. Moreover, we provide additional examples of models and of commonly used inter-event time densities. We also compare our method with an additional baseline.

Generally speaking, this work extends the idea of rejection-based simulation to networked systems that admit non-Markovian behavior. We propose RED, a rejection-based, event-driven simulation approach. RED is based on three main ideas:

1. We express the distributions of inter-event times as time-varying instantaneous rates (referred to as *intensity* or *rate functions*).

2. We sample inter-event times based on an over-approximation of the intensity function, which we counter-balance by using a rejection step.

3. We utilize a priority (resp. event) queue to decide which agent fires next.

The combination of these ideas allows to reduce the computational costs of each simulation step. More precisely, if an agent transitions from one local state to another one, no update of neighboring agents is required, even though their instantaneous rates might change as a result of the event. In short, the reason that it is not necessary to update an agent if its neighborhood changes, is that (by using the rate over-approximation) we always assume the "worst-case" behavior of an agent. If a neighboring agent is updated, the (actual) instantaneous rate of an agent might change but it will never exceed the rate over-approximation, which was used to

sample the firing time. Hence, the sampled firing time is always an under-approximation of the true one, regardless of what happens to adjacent nodes.

Naturally, this comes with a cost, in our case rejection (or null) events. Rejection events counter-balance the over-approximation of the instantaneous rate. The larger the difference between actual rate and over-approximated rate, the more rejection events will happen. Rejection events and the over-approximated rates complement each other and, in combination, yield a statistically correct (i.e. exact) algorithm. Utilizing a priority queue to order prospective events, renders the computational costs of each rejection step extremely small. We provide numerical results showing the efficiency of our method. In particular, we investigate how the runtime of our methods scales with the size and the connectivity of the contact network.

## Multi-agent model

ChangeColor Here we introduce our formalism for agent-based dynamics on complex networks. Our goal is to have a framework that is as expressive as possible while remaining intuitive. In particular, we make the following assumptions:

- The state transitions of an agent can depend on its whole (direct) neighborhood in a non-linear way.

- The time delay until an agent changes its state and the choice of a successor state can follow an arbitrary probability distribution whose parameterization depends on the agents' neighborhood.

- The number of local states can be arbitrarily high so that they are expressive enough to encode all information of interest.

- Individual nodes and edges may carry specific information that the model may take into account (e.g., some agents might be more reactive than others or connections might vary in strength).

With the above assumptions, it is possible to describe a wide range of epidemic-type applications (`SIS`, `SIR(S)`, threshold and voter model, etc.) as well as inter-event times following arbitrary distributions. We will also ensure that the framework is easily adaptable (e.g. to directed or temporal networks).

Next, we specify how, at any given point in time, a (global) *network state* is defined. After that, we explain how the dynamics can be formalized, that is, how agents change under the influence of two functions: $\phi$, for choosing a time of an agent's state change, and $\psi$, for choosing a local successor state. Note that, instead of explicitly using a probability density function (PDF) to encode the firing times of agents, we formulate so-called *intensity functions*. They have the same expressiveness but are more intuitive to use and easier to parametrize on the neighborhood of an agent. An intensity function determines how likely an agent fires in the following infinitesimal time interval. We will discuss in detail intensity functions in a dedicated section below.

Let $\mathcal{G} = (\mathcal{V}, \mathcal{E})$ be an undirected finite graph called *contact network* specifying the connectivity of the system. We assume that $G$ is strongly connected. That is, all nodes are reachable from all other nodes. Each node $v \in \mathcal{V}$ is an *agent*.

### Network state

At any given time point, the current (global) state of a network $\mathcal{G}$ is described by two functions:

- $S : \mathcal{V} \to \mathcal{S}$ assigns to each agent $v$ a local state $S(v) \in \mathcal{S}$, where $\mathcal{S}$ is a finite set of local states (e.g., $\mathcal{S} = \{\texttt{S}, \texttt{I}\}$ for the SIS model);

- $R : \mathcal{V} \to \mathbb{R}_{\geq 0}$, describes the residence time of each agent (the time elapsed since the agent changed its local state the last time).

We say an agent *fires* when it transitions from one local state to another. The time between two firings of an agent is denoted as *inter-event* time. Moreover, we refer to the remaining time until it fires as its *time delay*. The firing time depends on the direct neighborhood of an agent. At any point in time, the *neighborhood state* $M(v)$ of an agent $v$ is a set of triplets containing the local states and residence times of all neighboring agents and the agents themselves:

$$M(v) = \{(S(v'), R(v'), v') \mid (v, v') \in \mathcal{E}\} .$$

We use $\mathcal{M}$ to denote the set of all possible neighborhood-states in a given model.

## Network dynamics

Next, we specify how the network state evolves over time. Therefore, we assign to each agent $v \in \mathcal{V}$ two functions $\phi_v$ and $\psi_v$. $\phi_v$ governs when $v$ fires and $\psi_v$ defines its successor state. Both functions depend on the current state (first parameter), the residence time of of $v$ (second parameter), and the neighborhood $M(v)$ (third parameter).

- $\phi_v : \mathcal{S} \times \mathbb{R}_{\geq 0} \times \mathcal{M} \to \mathbb{R}_{\geq 0}$ defines the *instantaneous rate* of $v$. If $\lambda = \phi_v(S(v), R(v), M(v))$, then the probability that $v$ fires in the next infinitesimal time interval $t_\Delta$ is $\lambda t_\Delta$ (assuming $t_\Delta \to 0$);

- $\psi_v : \mathcal{S} \times \mathbb{R}_{\geq 0} \times \mathcal{M} \to P_\mathcal{S}$ determines the successor state when a transition occurs. More precisely, it determines for each local state the probability to be the successor state. Here, $P_\mathcal{S}$ denotes the set of all probability distributions over $\mathcal{S}$. Hence, if $p = \psi_v(S(v), R(v), M(v))$, then —assuming agent $v$ fires—the subsequent local state of $v$ is $s \in \mathcal{S}$ with probability $p(s)$.

Note that we assume that these functions have no pathological behavior. That is, we exclude the cases in which $\phi_v$ is defined so that it is not integrable or where some intensity function $\phi_v$ would cause an infinite amount of simulation steps in finite time (see Examples).

A multi-agent network model is fully specified by a tuple $(\mathcal{G}, \mathcal{S}, \{\phi_v\}, \{\psi_v\}, S_0)$, where $S_0$ denotes a function that assigns to each agent its initial local state.

## Examples

**Standard Markovian SIS model.** Consider the classical $\texttt{SIS}$ model with $\mathcal{S} = \{\texttt{S}, \texttt{I}\}$. $\phi_v$ and $\psi_v$ are the same for all agents:

$$\phi_v(s, t, m) = \begin{cases} c_r & \text{if } s = \texttt{I} \\ c_i \sum_{s', t', v' \in m} 1_{\texttt{I}}(s') & \text{if } s = \texttt{S} \end{cases} \qquad \psi_v(s, t, m) = \begin{cases} 1_{\texttt{S}} & \text{if } s = \texttt{I} \\ 1_{\texttt{I}} & \text{if } s = \texttt{S} \end{cases}$$

Here, $c_i, c_r \in \mathbb{R}_{\geq 0}$ are the infection and recovery rate constants. The infection rate is proportional to the number of infected neighbors while the recovery rate is independent from the neighborhood. Moreover, $1_s : \mathcal{S} \to \{0, 1\}$ is such that $1_s(s')$ is one if $s = s'$ and zero otherwise. The model is Markovian (w.r.t. all local states) as neither $\phi_v$ nor $\psi_v$ depends on the residence time of any agent. As in most binary state models, $\psi_v$ is deterministic in the sense that an agent in state $\texttt{I}$ always transforms to $\texttt{S}$ with probability one and vice versa.

**Complex cascade model.** ChangeColor Consider a modification of the independent cascade model [27] with local states susceptible (S), infected (I), and immune/removed (R). Infected nodes try to infect their susceptible neighbors. The infection attempts can be successful (turning the neighbor from S to I) or unsuccessful (turning the neighbor from S to R). Agents that are infected or immune remain in these states.

This model can be used to describe the propagation of some type of information on social media. Infected agents can be seen as having shared (e.g. re-tweeted) the information. A user exposed to the information decides right away if she considers it to be worth sharing. Seeing the information multiple times or from multiple sources does not increase the chances of sharing it (i.e. becoming infected). However, multiple infected neighbors might decrease the time until the information is perceived by an agent. Again, $\phi_v$ and $\psi_v$ are the same for all agents. We define the instantaneous rate

$$\phi_v(s, t, m) = \begin{cases} e^{-t_{\text{dist}}} & \text{if } s = \text{S and } \sum_{s', t', v' \in m} 1_{\text{I}}(s') > 0 \\ 0 & \text{otherwise.} \end{cases}$$

Here, $t_{\text{dist}}$ denotes the time elapsed since the latest infected neighbor became infected. Thus, the intensity at which infected agents "attack" their neighbors decreases exponentially and only the most recently infected neighbor counts. Moreover, the next internal state of an agent is selected according to the distribution:

$$\psi_v(s, t, m) = \{\text{I} \to p_i, \ \text{R} \to 1 - p_i, \ \text{S} \to 0\} \ ,$$

where $p_i \in [0, 1]$ denotes the infection probability. This example is both: non-Markovian, because the residence times of the neighbors influence the rate, and non-linear, because the individual effects from neighboring agents do not simply add up.

**Pathological behavior.** Assume two connected agents. Agent $A$ always stays in state S. Agent $B$ switches between states $\text{I}_1$ and $\text{I}_2$. The frequency at which $B$ alternates increases with the residence time of $A$ (denoted by $R(A)$). Let the rate to jump from $\text{I}_1$ to $\text{I}_2$ (and vice versa) be $\frac{1}{1-R(A)}$ for $R(A)<1$. Assume that we want to perform a simulation for the time interval $[0, 1]$. It is easy to see that the instantaneous rate of $B$ approaches infinity and the number of simulation steps (state transitions) will not converge. Generally speaking, pathological behavior may occur if $\phi_v(s, t, m)$ approaches infinity with growing $R(v')$ (within the simulation time), where $v'$ is a neighbor of $v$. However, it is allowed that $\phi_v(s, t, m)$ approaches infinity with growing $t$ ($t = R(v)$). Then no pathological behavior can occur because of the reset of $R(v)$ with each state transition of $v$.

## Semantics

We specify the semantics of a multi-agent model in a generative matter. That is, we describe a stochastic simulation algorithm that generates trajectories of the system. Recall that the network state is specified by the mappings $S$ and $R$. Let $t_{\text{global}}$ denote the global simulation time.

The simulation is based on a race condition among all agents: each agent picks a random firing time, but only the one with the shortest time delay wins and actually fires. Because each agent only generates a potential/proposed time delay, we might refer to this sampled value as *time delay candidate*. The algorithm starts by initializing the global clock $t_{global} = 0$ and setting $R(v) = 0$ for all $v \in \mathcal{V}$. The algorithm performs simulation steps until a predefined time horizon is reached. Each simulation step contains the following sub-steps:

1. Generate a random time delay candidate $t_v$ for each agent $v$. Identify the agent $v'$ with the smallest time delay $t_{v'}$.

2. Select the successor state $s'$ for $v'$ using $\psi_{v'}(S(v'), R(v') + t_{v'}, M(v'))$ and set $S(v') = s'$. Set $R(v') = 0$ and $R(v) = R(v) + t_{v'}$ for all $v \neq v'$.

3. Set $t_{global} = t_{global} + t_{v'}$ and go to Line 1.

This simulation approach is however—while being intuitive—very inefficient. Our approach, RED, will be statistically equivalent while being much faster.

**Generating time delays.** Recall that, in this manuscript, we encode inter-event time distributions using intensity functions. The intensity function of agent $v$ is used to generate time delay candidates which then compete in a race-condition (the shortest time delay "wins"). The relationship between time delays and intensities is further discussed in the next section.

There are several ways to generate a time delay candidate for an agent $v$. In one way or another, we have to sample from an exponential distribution with a time-varying rate parameter. In principle, there are many different possible methods for this. For an overview, we refer to [28–31].

An obvious way is to turn the intensity function induced by $\phi_v$ into a PDF (cf. Fig 1) and sample from it using inverse transform sampling. A more direct way is to perform numerical integration on $\phi_v$ assuming the neighborhood of $v$ stays the same. Let us, therefore, define for each $v$ the *effective rate* $\lambda_v(\cdot)$ which is the evolution of the intensity $\phi_v$ starting from the current time point, assuming no changes in the neighboring agents:

$$\lambda_v(t_\Delta) = \phi_n(S(n), R(n) + t_\Delta, M_{t_\Delta}(n)),$$
$$\text{where } M_{t_\Delta}(n) = \{(S(n'), R(n') + t_\Delta) \mid n, n' \in \mathcal{E}\} .$$

Here, $t_\Delta$ denotes the time increase of the algorithm.

The effective rate makes it possible to sample the time delay $t_v$ after which agent $v$ fires (if it wins the race), using the inversion transform method. First, we sample an exponentially distributed random variate $x$ with rate 1, then we integrate $\lambda_v(\cdot)$ to find $t_v$. Formally, $t_v$ is chosen such that the equation

$$\int_0^{t_v} \lambda_v(t_\Delta) dt_\Delta = x \tag{1}$$

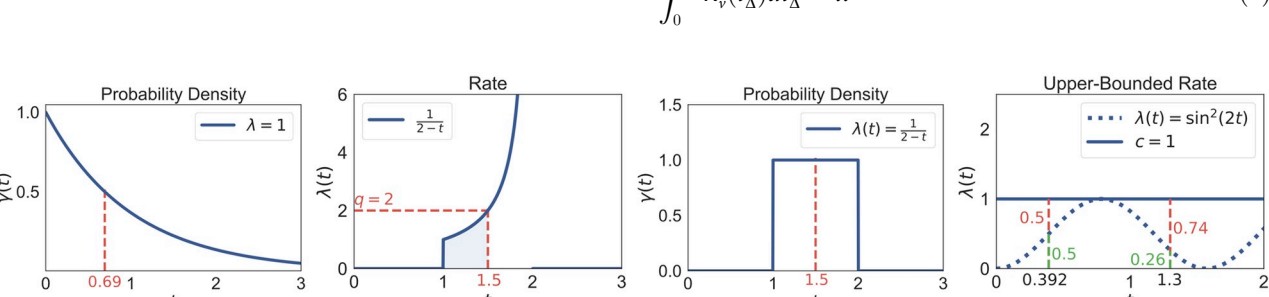

(a)                    (b)                    (c)                    (d)

**Fig 1. (a-c) Sampling event times with an intensity function** $\frac{1_{t \in [1,2]}}{2-t}$. (a) Generate a random variate from the exponential distribution with rate $\lambda = 1$, the sample here is 0.69. (b) We integrate the intensity function until the area is 0.69, here $t_n = 1.5$. (c) This is the intensity function corresponding to the uniform distribution in $\gamma(t) = 1_{t \in [1,2]}$. (d) Rejection sampling example: Sampling $t_v$ from a time-varying intensity function $\lambda(t) = \sin^2(2t)$ using an upper-bound of $c = 1$. Two iterations are shown with rejection probabilities shown in red. After one rejection step, the method accepts in the second iteration and returns $t_v = 1.3$.

is satisfied. The idea is the following: We first sample a random variate $x$ assuming a fixed rate (intensity function) of 1. The corresponding density is $\exp(-x)$, leading to $P(X > x) = \exp(-x)$ (sometimes referred to as *survival function*). Next, we consider the "real" time-varying intensity function $\lambda_v(\cdot)$ and choose $[0, t_v]$ such that the area under the time-varying intensity function is equal to $x$ (cf. Eq (1)). Hence,

$$P(X > x) = \exp(-x) = \exp\left(-\int_0^{t_v} \lambda_v(t_\Delta)dt_\Delta\right) = P(Y > t_v)$$

and $t_v$ is thus distributed according to the time-varying rate $\lambda_v(\cdot)$. Intuitively, by sampling the integral, we apriori define the number of infinitesimal time-steps we take until the agent will eventually fire. This number naturally depends on the rate function. If the rate decreases, more steps will be taken. We refer the reader to [29] for a proof.

An alternative approach to sample time delays is to use rejection sampling (this is not the rejection sampling which is the key of the RED method though) which is illustrated in Fig 1d. Assume that we have $c \in \mathbb{R}_{\geq 0}$ with $\lambda_v(t_\Delta) \leq c$ for all $t_\Delta$. We start with $t_v = 0$. Next, we sample a random variate $t'_v$ which is exponentially distributed with rate $c$. Next, we set $t_v = t_v + t'_v$ and accept $t_v$ with probability $\frac{\lambda_v(t_v)}{c}$. Otherwise, we reject $t'_v$ and repeat the process. If a reasonably tight over-approximation can be found, rejection sampling is much faster than numerical integration. The correctness can be shown similarly to the correctness of RED. That is, one creates a *complementing-* (or *shadow*-process) which accounts for the difference between the upperbound $c$ and $\lambda(t)$. In total, the null events and the complementing process cancel out, yielding statistically correct samples of $t_v$.

## Intensities and inter-event times

In our framework, the distribution of inter-event times is expressed using intensity functions. This is advantageous for the combination with rejection sampling. Here, we want to further establish the relationship between intensity functions and probability densities. Let us assume that at a given time point and for an agent $v$, the probability density that the agent fires after exactly $t_\Delta \in \mathbb{R}_{\geq 0}$ time units is given by a PDF $\gamma(t_\Delta)$. Leveraging the theory of renewal processes [31–33], we find the relationship

$$\lambda(t_\Delta) = \frac{\gamma(t_\Delta)}{1 - \int_0^{t_\Delta} \gamma(t')dt'} \qquad \text{and} \qquad \gamma(t_\Delta) = \lambda(t_\Delta)e^{-\int_0^{t_\Delta} \lambda(y)dy} \, .$$

We set $\lambda(t_\Delta)$ to zero if the denominator is zero. Using this equation, we can derive intensity functions from any given inter-event time distribution (e.g., uniform, log-normal, gamma, power-law, etc.). In cases where it is not possible to derive $\lambda(\cdot)$ analytically, we can still compute it numerically. Some examples of $\lambda(\cdot)$ for common PDFs are shown in Fig 2.

All density functions of time delays can be expressed as time-varying rates (i.e. intensities). However, only intensity functions with an infinite integral can be expressed as PDF. If $\int_0^\infty \lambda(t)dt$ is finite, the process might, with positive probability, not fire at all. This follows directly from Eq 1. The sampled reference area $x$ can be arbitrarily large, if it is larger than $\int_0^\infty \lambda(t)dt$ the process does not fire. For instance, consider an intensity function $\lambda(t)$ which is 1 if $t \in [0, 1]$ and zero otherwise. If $x > 1$, the process reaches $t = 1$ (without having already fired) and will also not fire while $t > 1$.

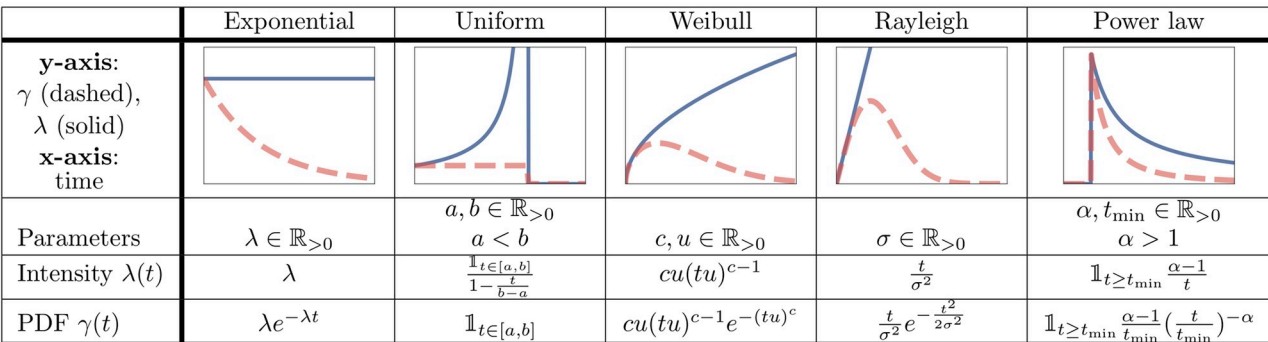

| | Exponential | Uniform | Weibull | Rayleigh | Power law |
|---|---|---|---|---|---|
| **y-axis**: $\gamma$ (dashed), $\lambda$ (solid) **x-axis**: time | | | | | |
| Parameters | $\lambda \in \mathbb{R}_{>0}$ | $a, b \in \mathbb{R}_{>0}$ $a < b$ | $c, u \in \mathbb{R}_{>0}$ | $\sigma \in \mathbb{R}_{>0}$ | $\alpha, t_{\min} \in \mathbb{R}_{>0}$ $\alpha > 1$ |
| Intensity $\lambda(t)$ | $\lambda$ | $\frac{\mathbb{1}_{t \in [a,b]}}{1 - \frac{t}{b-a}}$ | $cu(tu)^{c-1}$ | $\frac{t}{\sigma^2}$ | $\mathbb{1}_{t \geq t_{\min}} \frac{\alpha-1}{t}$ |
| PDF $\gamma(t)$ | $\lambda e^{-\lambda t}$ | $\mathbb{1}_{t \in [a,b]}$ | $cu(tu)^{c-1} e^{-(tu)^c}$ | $\frac{t}{\sigma^2} e^{-\frac{t^2}{2\sigma^2}}$ | $\mathbb{1}_{t \geq t_{\min}} \frac{\alpha-1}{t_{\min}} \left(\frac{t}{t_{\min}}\right)^{-\alpha}$ |

**Fig 2. Schematic illustration of intensity functions and inter-event time densities.** Examples of the relationship between common PDFs for inter-event times and their corresponding intensity functions. All functions are only defined on $t \geq 0$.

## Previous simulation approaches

Most recent work on non-Markovian dynamics focuses on formal models of such processes and their analysis [34–38]. Research has mostly focused on how specific distributions (e.g. uniformly distributed curing times) alter the behavior of the epidemic spreading, for instance, the epidemic threshold (see [3, 4] for an overview). Most of this work is, however, rather limited in scope, in the sense that only certain distributions or only networks with infinite nodes or only the epidemic threshold but not the full emerging dynamics are considered. Even though substantial effort was dedicated to the usage of rejection sampling in the context of Markovian stochastic processes on networks [24, 25, 39], only a few approaches are known to us that are dedicated to non-Markovian dynamics [22, 23].

Here, we shortly summarize the relevant algorithms in order to lay the grounds for our RED algorithm which was first introduced in [26]. We present an adaptation of the classical Gillespie method for networked processes as well as the non-Markovian Gillespie algorithm (nMGA) and its adaptation, the Laplace-Gillespie algorithm (LGA). To keep this contribution focused, we discuss all algorithms only for use in networked systems and within the notation of this paper.

## Non-Markovian Gillespie algorithm

Boguñá et al. develop a modification of the Gillespie algorithm for non-Markovian systems, nMGA [23]. Their method is statistically exact but computationally expensive. Conceptually, nMGA is similar to the baseline in Section Semantics but computes the time delays using so-called *survival functions* which simplifies the computation of the minimal time delay over all agents. An agent's survival function describes the probability that the time until its firing is larger than a certain threshold $t$ (for all $t$). The joint survival function of all agents determines the probability that all time delays are larger than $t$. The joint survival function can then be used to sample the next event time.

Unfortunately, in nMGA, it is necessary to iterate over all agents in each simulation step in order to construct the joint survival function. The authors also propose a fast approximation. Therefore, only the current instantaneous rate (at the beginning of each step) is used, and one assumes that all instantaneous rates remain constant until the next event. This is reasonable when the number of agents is very high because, if the number of agents approaches infinity, the time delay of the fastest agent approaches zero.

## Laplace-Gillespie algorithm

Masuda and Rocha have introduced the *Laplace-Gillespie algorithm* (LGA) in [22]. The method aims at reducing the computational costs of finding the next event time compared to nMGA. They only consider inter-event time densities that can be expressed as a continuous mixture of exponentials:

$$\gamma_v(t_\Delta) = \int_0^\infty p_v(\lambda)\lambda e^{-\lambda t_\Delta}d\lambda \ . \tag{2}$$

Here, $p_v$ is a PDF over the rate $\lambda_v \in \mathbb{R}_{\geq 0}$. The restriction of inter-event times limits the scope of the method to survival functions which are *completely monotone* [22]. The advantage is that we can sample the time delay $t_v$ of an agent $v$ by first sampling $\lambda_v$ according to $p_v$ and then sampling from an exponential distribution with rate $\lambda_v$. That is, $t_v = -\ln u/\lambda$ using for a uniformly (in $(0, 1)$) distributed random variate $u$.

## Our method

In this section, we propose the RED algorithm for the generation of statistically correct trajectories of non-Markovian spreading models on networks. The main idea is to use rejection sampling to reduce the computational cost of each simulation step. Specifically, when an agent changes its local state, we make it obsolete to update the rates of the agent's neighbors.

### Rate over-approximation

Recall that we use the effective rate $\lambda_v(\cdot)$ to express how the instantaneous rate of $v$ changes over time, assuming that no neighboring agent changes its state (colloquially, we extrapolate the rate into the future). A key ingredient of our approach is the construction of $\hat{\lambda}_v(\cdot)$ which upper-bounds the instantaneous rate of $v$, taking into consideration all possible state changes of $v$'s neighboring agents. That is, at all times $\hat{\lambda}_v(t_\Delta)$ is larger than (or equal to) $\lambda_v(t_\Delta)$ while we allow that arbitrary state changes of neighbors occur at arbitrary times in the future. In other words, $\hat{\lambda}_v(\cdot)$ upper-bounds $\lambda_v(\cdot)$ even when we have to re-compute $\lambda_v(\cdot)$ due to a changing neighborhood. Formally, the upper-bound always satisfies:

$$\hat{\lambda}_v(t_\Delta) \geq \sup_{M' \in \mathcal{M}_{v,t_\Delta}} \phi_n(S(n), R(n) + t_\Delta, M'), \tag{3}$$

where $\mathcal{M}_{v,t_\Delta}$ denotes the set of reachable neighborhoods (that is, with positive probability) of agent $v$ after $t_\Delta$ time units. Sometimes $\hat{\lambda}_v(\cdot)$ is referred to as a *dominator* of $\lambda_v(\cdot)$ [30].

Note that it is not feasible to compute the over-approximation algorithmically, so we derive it analytically. Upper-bounds can be constant or dependent on time. For multi-agent models (with a finite number of local states) time-dependent upper-bound exists for all practically relevant intensity functions since we can derive the maximal instantaneous rate w.r.t. all reachable neighborhood states which is typically finite except for some pathological cases (cf. Section Limitations).

**Example.** How does one find an over-approximation and why does it eliminate the need to update an agent's neighborhood? Consider again the Markovian SIS example from earlier. The recovery of an infected agent does not depend on its neighborhood. Hence, the rate is always $c_r$, which is also a trivial upper-bound. The rate at which a susceptible agent becomes infected is given by the $c_i$ times "number of infected neighbors". This means that the instantaneous infection rate of $v$ can be bounded by $\hat{\lambda}_v(t_\Delta) = k_v c_i$ where $k_v$ is the degree (number of

neighbors) of $v$. Note that this upper-bound does not depend on $t_\Delta$. When we use this upper-bound to sample the time delay candidate of an agent, this time point will always be an under-approximation. When a neighbor changes (e.g., becomes infected) the under-approximation remains valid.

However, consider for instance a recovery time that is uniformly distributed on [1, 2]. In this case, $\lambda_v(\cdot)$ approaches infinity (cf. Fig 1b) making a constant upper-bound impossible (even without considering any changes in the neighborhood).

## The **RED** algorithm

As input, our algorithm takes a multi-agent model $(\mathcal{G}, \mathcal{S}, \{\phi_v\}, \{\psi_v\}, S_0)$ and corresponding upper-bounds $\{\hat{\lambda}_v\}$. As output, the method produces statistically exact trajectories (samples) following the semantics introduced earlier. RED is based on two main data structures:

**Labeled graph.** A graph represents the contact network. In each simulation step, each agent (node) $v$ is annotated with its current state $S(v)$ and $T(v)$, the time point of its last state change.

**Event queue.** The event queue stores all (potential) future events (i.e. firings). An event is encoded as a tuple $(v, \hat{\mu}, \hat{t}_v)$, where $v$ is the agent that wants to fire, $\hat{t}_v$ the prospective absolute time point of firing, and $\hat{\mu} \in \mathbb{R}_{\geq 0}$ is an over-approximation of the true effective rate (at time point $\hat{t}_v$). The queue is sorted according to $\hat{t}_v$.

A global clock, $t_{\text{global}}$, keeps track of the elapsed time since the simulation started. We initialize the simulation by setting $t_{\text{global}} = 0$ and generating one event per agent. Using $T(v)$ (as in Line 2) is a viable alternative to using $R(v)$ in order to encode residence times since $R(v) = t_{\text{global}} - T(v)$. Practically $T(v)$ is more convenient, as it avoids the explicit updates of $R(v)$ for all agents after any event happens. Again, we simulate until some stopping criterion is fulfilled. Each simulation step contains the sub-steps:

1. Take the first event $(v, \hat{\mu}, \hat{t}_v)$ from the event queue and update $t_{\text{global}} = \hat{t}_v$.

2. Evaluate the true instantaneous rate $\mu = \phi_v(S(v), t_{\text{global}} - T(v), M(v))$ of $v$ at the current system state.

3. With probability $1 - \frac{\mu}{\hat{\mu}}$, **reject** the firing and go to Line 5.

4. Randomly choose the next state $s'$ of $v$ according to the distribution $\psi_v(S(v), t_{\text{global}} - T(v), M(v))$. If $S(v) \neq s'$: set $S(v) = s'$ and $T(v) = t_{\text{global}}$.

5. Generate a new event for agent $v$ and push it to the event queue.

6. Go to Line 1.

The main difference to previous approaches is that traditionally the rate has to be updated for all neighbors of a firing agent. In RED only the rate of the firing agent has to be updated.

**Event generation.** Here, we specify how the event generation in Line 5 is done. We sample a random time delay $t_v$ according to $\hat{\lambda}_v(\cdot)$ and set $\hat{t}_v = t_{\text{global}} + t_v$ (because the event contains the absolute time). To sample $t_v$ according to the over-approximated rate, we either use the numerical integration of Eq (1) or sample directly from an exponential distribution which upper-bounds the intensity function (cf. Fig 1d). Finally, we set $\hat{\mu} = \hat{\lambda}_v(t_v)$. Alternatively, when appropriate for $\hat{\lambda}_v(t)$, we can even use an LGA-like approach to sample $t_v$ (and also set $\hat{\mu} = \hat{\lambda}_v(t_v)$) [23].

## Asymptotic time complexity

ChangeColor Here, we discuss how the runtime of RED scales with the size of the underlying contact network (and the number of agents). Assume that a binary heap is used to implement the event queue and that the graph structure is implemented using a hashmap. Each step starts by popping an element from the queue which has constant time complexity. Next, we compute $\mu$. Therefore, we have to look up all neighbors of $v$ in the graph structure and iterate over them. We also have to look up all states and residence times. This step has linear time-complexity in the number of neighbors. More precisely, lookups in the hashmaps have constant time-complexity on average and are linear in the number of agents in the worst case. Computing the rejection probability has constant time complexity. When no rejection events takes place, we update $S$ and $T$. Again, this has constant time-complexity on average. Generating a new event does not depend on the neighborhood of an agent and has, therefore, constant time-complexity. Note that this step can still be somewhat expensive when it requires integration to sample $t_e$ but not in an asymptotic sense. Thus, a step in the simulation is linear in the number of neighbors of the agent under consideration.

In contrast, previous methods require that after each update, the rate of each neighbor $v'$ is re-computed. The rate of $v'$, however, depends on the whole neighborhood of $v'$. Hence, it is necessary to iterate over all neighbors $v''$ of every single neighbor $v'$ of $v$.

## Correctness

ChangeColor The correctness of RED can be shown similarly to [24]. Here, we provide a proof-sketch. First, consider the rejection-free version of the algorithm:

1. Take the first event $(v, \mu, \hat{t}_v)$ from the event queue and update $t_{\text{global}} = \hat{t}_v$.

2. Randomly choose the next state $s'$ of $v$ according to the distribution $\psi_v$ https://www.overleaf.com/project/5e26f009c28b95000160ad41 $(S(v), t_{\text{global}} - T(v), M(v))$.

3. If $S(v) = s'$: Generate a new event for agent $v$, push it to the event queue, and go to Line 1 (no state transition of $v$).

4. Otherwise set $S(v) = s'$ and generate a new event for agent $v$ and push it to the event queue.

5. For each neighbor $v'$ of $v$: Remove the event corresponding to $v'$ from the queue and generate a new event (taking the new state of $v$ into account).

6. Go to Line 1.

Rejection events are not necessary for this version of the algorithm because all events in the queue are generated by the "real" rate and are therefore consistent with the current system state. In other words, the rejection probability would always be zero. It is easy to see that the rejection-free version is a direct event-driven implementation of the naïve simulation algorithm which was introduced in the Semantics Section. The correspondence between Gillespie-approaches and event-driven simulations is exploited in the literature, for instance in [4]. Thus, it is sufficient to show that the above rejection-free simulation and RED are statistically equivalent.

First, note that it is possible to include *self-loop events* to our model without changing the underlying dynamics (resp. statistical properties). These are events in which an agent fires but transitions into the same internal state it already occupies. Until now, we did not allow such self-loop behavior. In the algorithm, self-loop events correspond to the condition $S(v) = s'$ in the third step. Such events do not alter the network state and, therefore, do not change the

statistical properties of the generated trajectories. The key idea is now to change $\phi_v$ and $\psi_v$ to $\hat{\phi}_v$ and $\hat{\psi}_v$, respectively, such that the events related to $\hat{\phi}_v$ and $\hat{\psi}_v$ also admit self-loop events with a certain probability. Specifically, self-loops have the same probability as rejection events in the RED method but, apart from that, $\hat{\phi}_v$ and $\hat{\psi}_v$ admit the same behavior as $\phi_v$ and $\psi_v$. Formally, this is achieved by using so-called *shadow-processes* [24, 39]; sometimes also referred to as *complementing process* [30]. A shadow-process does not change the state of the corresponding agent but still fires at a certain rate. In the end, we can interpret the rejection events not as rejections, but as the statistically necessary application of the shadow-process.

We define the rate of the shadow-process, denoted by $\tilde{\lambda}$, to be the difference between the rate over-approximation and the true rate. For all $v$, $t$, this gives rise to the invariance:

$$\hat{\lambda}_v(t) = \lambda_v(t) + \tilde{\lambda}_v(t) \ .$$

We define $\hat{\psi}_v$ such that it includes the shadow-process.

The only thing remaining is to define $\hat{\phi}_v$ such that the shadow-process does not influence the system state. Therefore, we simply trigger a *null event* (or self-loop) with the probability that is proportional to how much of $\hat{\phi}_v$ is induced by the shadow-process. Hence, the probability for a null event is $\frac{\tilde{\phi}_v}{\hat{\phi}_v}$. Consequently,

$$\hat{\psi}_v(s, t, m) = p \qquad \text{where } p \text{ is defined such that :}$$
$$p(s) = \frac{\tilde{\phi}_v}{\hat{\phi}_v}, \qquad p(s') = \left(1 - \frac{\tilde{\phi}_v}{\hat{\phi}_v}\right) \psi_v(s', t, m) \ \ (\forall s' \neq s)$$

W.l.o.g., we assume that the original system has no inherent self-loops. In summary, the model specification with and without the shadow-process are equivalent (i.e., admit the same dynamics). This is because it has no actual effect on the system, all the additional reactivity is compensated by the null event. Secondly, simulating the rejection-free algorithm including the shadow-process directly yields RED. In particular, the rejections events have the same likelihood as the shadow-process being chosen in $\hat{\psi}$.

## Limitations

The practical and theoretical applicability of our approach depends on how well the intensity function of an agent can be over-approximated. The larger the difference between $\lambda(\cdot)$ and $\hat{\lambda}(\cdot)$ becomes, the more rejection events occur and the slower our method becomes. In general, since rejection events are extremely cheap, it is not a problem for our method when most of the events in the event queue will be rejected.

However, it is easy to think of examples where RED will perform exceptionally bad. For instance, consider an SIS-type model, but nodes can only become infected if exactly half of their neighbors are infected. In this case, the over-approximation would assume that for all susceptible nodes this is always the case, causing too many rejection events. Likewise, the problem can also occur in the time domain. Consider the case that infected nodes only infect their susceptible neighbors in the first $t_\Delta$ time-units of their infection with rate $\lambda$, where $t_\Delta$ is extremely short (e.g. 0.001) and $\lambda$ is extremely high (e.g. 1000). Given a susceptible node, we do not know how many of its neighbors will be newly infected in the future, so we have to assume that all neighbors are infectious all of the time.

Similarly, in some cases, it might not be possible to find a theoretical upper-bound for the rate at all. Consider the case where an infected agent with residence time $t$ attacks its neighbors

at rate $|-\log(t)|$ (which converges to infinity for $t \to 0$). This still gives rise to a well-defined stochastic process because the integration of $|-\log(t)|$ leads to non-zero inter-event times and, therefore, it is possible to sample inter-event times even though the rate starts at infinity. However, we cannot build an upper-bound because, again, we have to assume that all neighbors of a susceptible node are always newly infected.

There are also more practical examples like networked (self-exiting) Hawkes processes [40]. Here, the number of firings of a neighbor increases the instantaneous rate of an agent. As it is not possible to bound (in advance) the number of times the neighbors fire (at least not without additional assumptions), it is not possible to construct an upper-bound for the intensity function for any future point in time.

## Case studies

We demonstrate the effectiveness of `RED` on three case studies. We generate synthetic graphs to use as contact networks. Therefore, we use the stochastic *configuration model* where the degree distribution is specified by a truncated power-law [41]. That is, for a degree $k$, $P(k) \propto k^{-\beta}$ for $3 \le k \le |\mathcal{N}|$. We use $\beta \in \{2, 2.5\}$ (a smaller value for $\beta$ leads to to a larger average degree and higher connectivity). `RED` is implemented in Julia and publicly available (`github.com/gerritgr/non-markovian-simulation`). The evaluation was performed on a 2017 MacBook Pro with a 3.1 GHz Intel Core i5 CPU. Runtime results for different models are shown in Fig 3. To compute the step-wise CPU time, we completely ignore the rejection steps to not give our method an advantage. We remark that `RED` and `Baseline` are both statistically correct, meaning that they sample from the correct distribution in the sense of the model semantics, while `nMGA` provides an approximation.

## Baseline

We compare the performance of `RED` with a baseline-algorithm and and an `nMGA`-type approach. As a baseline, we use the rejection-free variant of the algorithm where, when an agent fires, all of its neighbors are updated (described in more detail in Section Correctness). In the Voter-model the baseline uses an `LGA`-type approach to sample inter-event times (following Eq 2). In the other experiments we sample inter-event times using the rejection-based approach from Fig 1d. We do note that `LGA` and `RED` are not directly comparable as they are associated with different objectives. In short, `LGA` focuses on optimizing the generation of inter-event times while `RED` aims at reducing the number of times that are necessary to generate inter-event times. We want to emphasize that the reason we include an `LGA`-type and

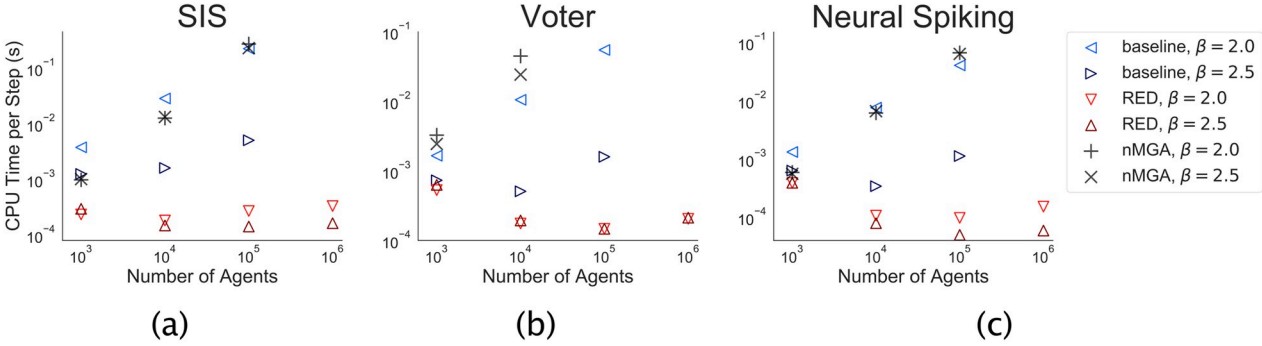

**Fig 3. Results.** Computation time of a single simulation step w.r.t. network size and connectivity (smaller $\beta \Rightarrow$ higher connectivity). We measure the CPU time per simulating step by dividing the simulation time by the number of successful (i.e., non-rejection) steps.

nMGA-type sampling approach is to highlight that our performance gain is not part of the specifics on how inter-event times are generated.

We use an nMGA-type method as a second comparison. It is a re-implementation of the approximate version of nMGA from [23]. The method stores all agents with their associated residence time in a list. In each step, we iterate over the list and generate a new firing time (candidate) for each agent assuming that the instantaneous rate remains constant (note that assuming a constant rate means sampling an exponentially distributed time delay). Then, the agent with the shortest time delay candidate fires and the residence times of all agents are updated. The approximation error decreases with an increasing network size because the time periods between events become smaller.

## SIS model

As our first test case, we use a non-Markovian modification of the classical SIS model. Specifically, we assume that infected nodes become (exponentially) less infectious over time. That is, the rate at which an infected agent with residence time $t$ attacks its susceptible neighbors is $ue^{-ut}$ for $u = 0.4$. This does not directly relate to a probability density because the infection event might not happen at all. Empirically, we choose parameters which ensure that the infection actually spreads over the network. We upper-bound the rate at which a susceptible agent $v$ (with degree $k_v$) gets infected with $\hat{\lambda}_v(t) = uk_v$. The upper-bound is constant in time and conceptually similar to the earlier example (cf. Section Rate over-approximation). We sample $t_v$ using an exponential distribution (i.e., without numerical integration). The time until an infected agent recovers is independent from its neighborhood and uniformly distributed in [0, 1] (similar to [42]). Hence, we can sample it directly. We start with 5% randomly selected infected agents.

## Voter model

The voter model describes the spread of two competing opinions, denoted as A and B. Agents in state A switch to B and vice versa, thus $\psi$ is deterministic.

In this experiment we use an inter-event time that can be sampled using an LGA-type approach (cf. Eq 2). Moreover, to take full advantage of the LGA-formulation, we assume that the neighborhood of an agent modulates the PDF $p_v$ which specifies the continuous mixture of rates (otherwise, we could simply pre-compute it). Here, we choose $p_v$ to be a uniform distribution in [0, $o_v$], where $o_v$ is the fraction of opposing neighbors of agent $v$. That is, if $v$ is in state A (resp. B), then $o_v$ is the number of neighbors in B (resp. A) divided by $v$'s degree $k_v$.

Hence, we can sample an time delay candidate by sampling a uniformly distributed random variate $\lambda_v \in [0, o_v]$ and then sampling the time delay candidate $t_v$ which is exponentially distributed with rate $\lambda_v$. The resulting inter-event time distribution resembles a power-law with a slight exponential cut-off [22]. The cut-off becomes more dominant for larger $o_v$. Formally,

$$\gamma_v(t) \quad = \int_0^{o_v} \frac{1}{o_v} \lambda e^{-\lambda t} d\lambda \quad = \frac{1 - e^{-o_v t}(1 + o_v t)}{o_v t^2} \quad \text{and}$$

$$\lambda_v(t) \quad = \frac{1}{t} - \frac{o_v}{e^{o_v t} - 1} \quad t \geq 0 \ .$$

To upper-bound the instantaneous rate we set $o_v = 1$. To sample $t_v$ in RED, we use rejection sampling (Fig 1d). The baseline uses the LGA-based approach, but changing to rejection sampling does not noticeably change the performance. We initialize the simulation with 50% of agents in A and B respectively.

## Neural spiking

ChangeColor To model neural spiking behavior, we propose a networked (i.e., multivariate) *temporal point-processes* [43]. In temporal point-processes, agents are naturally excitable (S) and can get activated for an infinitesimally short period of time (I). After that, they become immediately excitable again. Point-processes model scenarios where one is only interested in the firing times of an agent not in their local state. They are commonly used to model spiking behavior of neurons [44, 45] or information propagation in social networks (like re-tweeting) [40]. A random trajectory of a system identifies each agent $v$ with a list of time points $\mathcal{H}_v$ of its activations. Here, we consider multivariate point-processes, where each agent (node) represents one point-processes and neighboring agents influence each other by inhibition or excitement. Therefore, we identify each (undirected) edge $v$, $u$ with a weight $w_{v,u}$ of either 1 (excitatory connection) or −1 (inhibitory connection). Moreover, neurons can spontaneously fire with a baseline intensity of $\mu \in \mathbb{R}_{\geq 0}$. Formally,

$$\phi_v(s, t, m) \quad = f\left(\mu + \sum_{s', t', v' \in m} \frac{w_{v,v'}}{1 + t'}\right)$$

$$\text{with} \quad f(x) \quad = \max(0, \tanh(x)) \, .$$

The function $f$ is called a *response function*, it converts the synaptic input into the actual firing rate. We use the same one as in [46]. Without $f$, the intensity could become negative. Note that $\psi_v$ is deterministic. Our model can be seen as a non-Markovian modification of the model of Benayoun et al. in [46]. Contrary to Benayoun et al., we do not assume that active neurons stay in their active state for a specific (in their case, exponentially distributed) amount of time. Instead, we assume that they become immediately excitable again and that an activation affects the neighboring neurons through a kernel function $1/(1 + t')$. The kernel ensures that neighbors who fired more recently (i.e., have a smaller residence time $t'$) have a higher influence on an agent.

The residence time of an agent itself does not influence its rate. In contrast to multivariate self-exiting Hawkes processes, only the most recent firing—and not the whole event history $\mathcal{H}_v$—contributes to the intensity of neighboring agents [40, 47]. Taking the whole history into account is not easily possible with a finite amount of local states and introduces intensity functions which cannot be upper-bounded (cf. Limitations). For our experiments, we set $\mu = 0.01$, define 20% of the edges to be inhibitory and use the trivial upper-bound of one (induced by the response function).

**Discussion.** Our experimental results provide a clear indication that rejection-bases simulation (and the corresponding over-approximation of the instantaneous rate) can dramatically reduce the computational costs of stochastic simulation in the context of non-Markovian simulation on networks.

As expected, we see that the runtime behavior is influenced by the number of agents (nodes) and the number of interconnections (edges). Interestingly, for RED, the number of edges seems to be much more relevant than the number of agents. Most noticeably, the CPU time of each simulation step does practically not increase (beyond statistical noise) with the number of nodes. Moreover, one can clearly see that RED consistently outperforms the baseline up to several orders of magnitude (cf. Fig 3), while the gain in computational time (i.e., RED CPU time by baseline CPU time) ranges from 10.2 ($10^3$ nodes, voter model, $\beta = 2.5$) to 674 ($10^5$ nodes, SIS model, $\beta = 2.0$).

Note that, we only compared an LGA-type sampling approach with our method in the voter model experiment The other case-studies could not straightforwardly be simulated with LGA

due to its constraints on the time delays. However, we still assume that the rejection-free base-line algorithm is comparable with `LGA` in the other experiments as both of them only update the rates of the relevant agents after an event. We also tested an `nMGA`-like implementation where rates are considered to remain constant until the next event. However, the method scales—albeit it is only approximate—worse than the baseline.

Note that the `SIS` model is somewhat unfavorable for `RED` as it leads to the generation of a large number of rejection events, especially when only a small fraction of agents are overall infected. For concreteness, consider an agent with many neighbors of which only very few are infected. The over-approximation simply assumes that *all* neighboring agents are infected *all the time*. Nevertheless, the low computational costs of each rejection event seem to easily atone for their large number. In contrast, the neural spiking model is very favorable for our method as the tanh(·) response function provides a global upper-bound for the instantaneous rate of each agent. Performance-wise the differences between the two models are, surprisingly, pretty small.

## Conclusion

We proposed `RED`, a rejection-based algorithm for the simulation of non-Markovian agent models on networks. The key advantage, and most significant contribution of our method, is that it is no longer required to update the instantaneous rates of the whole neighborhood in each simulation step. This practically and theoretically reduces the time complexity of each step compared to previous simulation approaches and makes our method viable for the simulation of dynamical processes on real-world networks which often have millions of nodes. In addition, rejection steps provide for some inter-event time distributions a fast alternative to integrating the intensity function. Currently, the most notable downside of the method is that the over-approximations $\hat{\lambda}$ have to be constructed manually. It remains to be determined if it is possible to automate the construction of $\hat{\lambda}$ in an efficient way as the trivial way of searching in the state space of all reachable neighborhoods is not feasible. We also plan to investigate how correlated events (as in [22, 48]) can be integrated into `RED`.

## Author Contributions

**Conceptualization:** Gerrit Großmann.

**Formal analysis:** Gerrit Großmann, Luca Bortolussi, Verena Wolf.

**Funding acquisition:** Verena Wolf.

**Investigation:** Luca Bortolussi, Verena Wolf.

**Methodology:** Gerrit Großmann, Luca Bortolussi, Verena Wolf.

**Project administration:** Gerrit Großmann, Luca Bortolussi, Verena Wolf.

**Resources:** Gerrit Großmann.

**Software:** Gerrit Großmann.

**Supervision:** Gerrit Großmann, Luca Bortolussi, Verena Wolf.

**Validation:** Gerrit Großmann.

**Visualization:** Gerrit Großmann.

**Writing – original draft:** Gerrit Großmann, Luca Bortolussi, Verena Wolf.

**Writing – review & editing:** Gerrit Großmann, Luca Bortolussi, Verena Wolf.

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
