## [Decision Letter · Decision Letter 0]

22 Apr 2020

PONE-D-20-05820

Efficient simulation of non-Markovian dynamics on complex networks

PLOS ONE

Dear Mr. Grossmann,

Thank you for submitting your manuscript to PLOS ONE. After careful consideration, we feel that it has merit but does not fully meet PLOS ONE’s publication criteria as it currently stands. Therefore, we invite you to submit a revised version of the manuscript that addresses the points raised during the review process.

We would appreciate receiving your revised manuscript by Jun 05 2020 11:59PM. To enhance the reproducibility of your results, we recommend that if applicable you deposit your laboratory protocols in protocols.io, where a protocol can be assigned its own identifier (DOI) such that it can be cited independently in the future. For instructions see: http://journals.plos.org/plosone/s/submission-guidelines#loc-laboratory-protocols

We look forward to receiving your revised manuscript.

Kind regards,

Hocine Cherifi

Academic Editor

PLOS ONE

Journal Requirements:

Großmann G., Bortolussi L., Wolf V. (2020) Rejection-Based Simulation of Non-Markovian Agents on Complex Networks. In: Cherifi H., Gaito S., Mendes J., Moro E., Rocha L. (eds) Complex Networks and Their Applications VIII. COMPLEX NETWORKS 2019. Studies in Computational Intelligence, vol 881. Springer, Cham

I appreciate that this is your own conference paper which this manuscript is an extension of, and that may you have revised the text substantially already for this submission, but please ensure that as much of the overlapping text as possible is revised or removed, in particular as the previous text is under copyright.   

'This work was partly funded by the German Research Foundation (DFG) via the collaborative research center “Methods and Tools for Understanding and Controlling Privacy"'

a. Please provide an amended statement that declares *all* the funding or sources of support (whether external or internal to your organization) received during this study, as detailed online in our guide for authors at http://journals.plos.org/plosone/s/submit-now 

Please also include the statement “There was no additional external funding received for this study.” in your updated Funding Statement.

'No'

a. Please complete your Competing Interests statement to state any Competing Interests. If you have no competing interests, please state "The authors have declared that no competing interests exist.", as detailed online in our guide for authors at http://journals.plos.org/plosone/s/submit-now

Reviewers' comments:

Reviewer's Responses to Questions

**Comments to the Author**

1. Is the manuscript technically sound, and do the data support the conclusions?

Reviewer #1: Yes

Reviewer #2: Yes

Reviewer #3: Yes

2. Has the statistical analysis been performed appropriately and rigorously? 

Reviewer #1: Yes

Reviewer #2: N/A

Reviewer #3: I Don't Know

3. Have the authors made all data underlying the findings in their manuscript fully available?

Reviewer #1: Yes

Reviewer #2: Yes

Reviewer #3: Yes

4. Is the manuscript presented in an intelligible fashion and written in standard English?

Reviewer #1: Yes

Reviewer #2: Yes

Reviewer #3: Yes

5. Review Comments to the Author

Reviewer #1: Review for Efficient simulation of non-Markovian dynamics on complex network to Plos One.

Authors propose a stochastic rejection-based, event-driven simulation algorithm

that scales well with the size of a network and connectivity of the underlying contact network.

Authors check that the method produces statistically correct samples.

The paper addresses interesting question about what are the alternatives

to use stochastic (Monte-Carlo) simulations to study the complex emerging dynamical patterns.

At the same time there are several suggestions I would like to make about the paper:

1. In the description of the multi-agent model

authors are describing that model assumptions can describe wide range of applications,

they also briefly mention "as well as non-Markovian delays." (line 80=)

I would recommend either to put a reference here on what exactly they mean

or to elaborate here on this point, since it seems

it could be understood in multiple ways.

I understand that this work is is an extension of [26] in terms of theoretical analysis, at the same

time it feels that it needs to be more clear for those who did not read that reference.

2. for making the point of the proposed method clear I would suggest to include a schme or a diagram

to Network dynamics description on page 10. It would clarify to readers

about what is the role of \\psi and \\phi functions.

3. in line 92 when authors refer to connectivity of a network

it is not clear whether they refer to a network to be fully connected component,

so I would specify it there

4. line 104 type with "of"

5. in conclussions I would add eleborations on comparing methods to some other methods -

when it is performing comparably well.

6. from 164 line authors describe delay generation, i would again suggest to give more links to other

7. in equation between lines 350 and 351 it would be better to specify p(s) instead of p

8. in caption of fig.2 i would specify more what is in the brackets [smaller is better].

in general for the RED evaluation it would be useful to give readers some guidelines in which cases the

method performs well and in which it is not as effective (when the update the instantaneous rates of the whole 473

neighborhood in each simulation step would drastically decrease optimality to use other algorithms)

Overall, I recommend minor revisions with keeping in mind making the whole paper to explain

the main methods in more intuitive way.

Thank you.

Reviewer #2: The manuscript "Efficient simulation of non-Markovian dynamics on complex networks" by Gerrit Grossmann et al. presents a new method for simulating continuous time non-Markovian dynamical processes on static networks.

Using a delay time generated according to an overestimated instantaneous rate followed by a rejection-based selection permits to avoid updating the rates of neighboring nodes and allows to simulate processes in a highly efficient way.

The proposed method seems very interesting and promising. However, I believe that several parts of the manuscript could be made clearer and that relevant references should be added.

Concerning the case studies, it would be interesting to test the performances with existing algorithms and to see that the results obtained are similar (in the statistical sense) to results obtained with other algorithms.

This would also add to this manuscript compared to the already published paper in the complex networks conference.

A part from this and several typos, I think the manuscript is clear and thorough.

More precisely, in the part about generating time-delays, the validity of the method of integration (Fig. 1 and eq. 1) should be demonstrated. Why is t_v distributed according to the correct distribution? It may be obvious to the authors, but a small justification would make it clearer to the readers. References should also be provided.

Also, Fig. 1d should be clearly explained in the main text and in the caption.

References for the rejection-based method should also be provided.

Reasoning in term of intensity instead of PDF is not intuitive for most readers, given that the authors chose to submit to a wide audience journal, it would be nice if they tried to help the readers to understand those concepts.

For example, when they say that if the integral of the intensity is finite, the process might not fire at all with positive probability, this is not obvious for me. Could they give an example or provide a reference?

Another problem is that they never actually clearly explain why, with their method, the rates of the neighbors of a firing agent do not need to be updated. This should be explained in the introduction and in the description of the method.

About the case studies, as said above, when proposing a new simulation methods it sounds normal to show how it compares to existing methods (as long as the codes for these methods is easily available).

Could the authors find a case study where LGA, nMGA and RED could be compared?

Also, it would be nice to see that the (ensemble average of the) results of the simulations agree.

There are also several typos:

- page 2, line 1: "that based" -> "based"

- line 18: "on models on so-called" -> "on so-called"

- page 3, line 85: "a an agent" -> "an agent"

- page 4, line 99: "the the agent" -> "the agent"

- page 5, line 137: "dented by" ?

- page 8, line 221: "only use the" -> "only the"

- page 11, line 339: "introduced the" -> "introduced in the"

- page 11, line 340: "simulationS" and there is no verb in this sentence.

- line 345/346: "process" -> "processes"

- Discussion: reference Fig. 2.

Reviewer #3: The manuscript by Großmann and coworkers presents an

alternative method for efficient simulation of non-Markovian

processes on networks for generic dynamical processes.

The method is grounded on optimized rejection algorithms.

The issue is very timely and methodology is apparently

sound. The codes in Julia are available in github.

The text is grammatically well written. However, the flow of

reading is not good since it is quite prolix. I think

that the manuscript could be much more concise allowing to

grab the important ideas more quickly. The number of didactic examples

is exaggerated and reading not efficient.

Another serious issue is that only CPU times with respect

to different methods were presented but no results showing the

accuracy of the optimized strategies were discussed. I think

that it is a bad balance of content where too much space is

devoted to pedagogic aims in detriment to discussion and results

on the accuracy.

So, I recommend a major revision shrinking the

introductory sections and expanding results and discussions.

6. PLOS authors have the option to publish the peer review history of their article (what does this mean?). If published, this will include your full peer review and any attached files.

Reviewer #1: No

Reviewer #2: No

Reviewer #3: No

---

## [Author Response · Author response to Decision Letter 0]

6 Oct 2020

The response to the reviewers is included in the submitted files.

---

## [Editor Report · Decision Letter 1]

14 Oct 2020

Efficient simulation of non-Markovian dynamics on complex networks

PONE-D-20-05820R1

Dear Dr. Grossmann,

We’re pleased to inform you that your manuscript has been judged scientifically suitable for publication and will be formally accepted for publication once it meets all outstanding technical requirements.

Kind regards,

Hocine Cherifi

Academic Editor

PLOS ONE

---

## [Editor Report · Acceptance letter]

16 Oct 2020

PONE-D-20-05820R1 

Efficient simulation of non-Markovian dynamics on complex network 

Dear Dr. Grossmann:

I'm pleased to inform you that your manuscript has been deemed suitable for publication in PLOS ONE. Congratulations! Your manuscript is now with our production department. 

Kind regards, 

on behalf of

Professor Hocine Cherifi 

Academic Editor

PLOS ONE